# XPF–ERCC1 Blocker Improves the Therapeutic Efficacy of 5-FU- and Oxaliplatin-Based Chemoradiotherapy in Colorectal Cancer

**DOI:** 10.3390/cells12111475

**Published:** 2023-05-25

**Authors:** Ming-Yii Huang, Yi-Jung Huang, Tian-Lu Cheng, Wun-Ya Jhang, Chien-Chih Ke, Yi-Ting Chen, Shih-Hsun Kuo, I-Ling Lin, Yu-Hsiang Huang, Chih-Hung Chuang

**Affiliations:** 1Department of Radiation Oncology, Kaohsiung Medical University Hospital, Kaohsiung Medical University, Kaohsiung 80708, Taiwan; 2Department of Radiation Oncology, School of Medicine, College of Medicine, Kaohsiung Medical University, Kaohsiung 80708, Taiwan; 3Center for Cancer Research, Kaohsiung Medical University, Kaohsiung 80708, Taiwan; 4Department of Biochemistry, School of Post Baccalaureate Medicine, College of Medicine, Kaohsiung Medical University, Kaohsiung 80708, Taiwan; 5Drug Development and Value Creation Research Center, Kaohsiung Medical University, Kaohsiung 80708, Taiwan; 6Department of Biomedical Science and Environmental Biology, Kaohsiung Medical University, Kaohsiung 80708, Taiwan; 7Department of Medical Laboratory Science and Biotechnology, Kaohsiung Medical University, Kaohsiung 80708, Taiwan; 8Department of Medical Imaging and Radiological Sciences, Kaohsiung Medical University, Kaohsiung 80708, Taiwan; 9Department of Medical Research, Kaohsiung Medical University Hospital, Kaohsiung 80708, Taiwan; 10Department of Pathology, Kaohsiung Medical University Hospital, Kaohsiung Medical University, Kaohsiung 80708, Taiwan; 11Department of Pathology, School of Medicine, College of Medicine, Kaohsiung Medical University, Kaohsiung 80708, Taiwan; 12Post-Graduate Year Training, Kaohsiung Medical University Hospital, Kaohsiung Medical University, Kaohsiung 80708, Taiwan

**Keywords:** 5-FU-based CRT, OXA-based CRT, ERCC1, XPF–ERCC1 blocker, colorectal cancer

## Abstract

5-FU-based chemoradiotherapy (CRT) and oxaliplatin-based CRT are commonly used therapies for advanced colorectal cancer (CRC). However, patients with a high expression of ERCC1 have a worse prognosis than those with a low expression. In this study, we investigated the effect of XPF–ERCC1 blockers on chemotherapy and 5-FU-based CRT and oxaliplatin (OXA)-based CRT in colorectal cancer cell lines. We investigated the half-maximal inhibitory concentration (IC_50_) of 5-FU, OXA, XPF–ERCC1 blocker, and XPF–ERCC1 blocker, and 5-FU or OXA combined and analyzed the effect of XPF–ERCC1 blocker on 5-FU-based CRT and oxaliplatin-based CRT. Furthermore, the expression of XPF and γ-H2AX in colorectal cells was analyzed. In animal models, we combined the XPF–ERCC1 blocker with 5-FU and OXA to investigate the effects of RC and finally combined the XPF–ERCC1 blocker with 5-FU- and oxaliplatin-based CRT. In the IC_50_ analysis of each compound, the cytotoxicity of the XPF–ERCC1 blocker was lower than that of 5-FU and OXA. In addition, the XPF–ERCC1 blocker combined with 5-FU or OXA enhanced the cytotoxicity of the chemotherapy drugs in colorectal cells. Furthermore, the XPF–ERCC1 blocker also increased the cytotoxicity of 5-FU-based CRT and OXA -based CRT by inhibiting the XPF product DNA locus. In vivo, the XPF–ERCC1 blocker was confirmed to enhance the therapeutic efficacy of 5-FU, OXA, 5-FU-based CRT, and OXA CRT. These findings show that XPF–ERCC1 blockers not only increase the toxicity of chemotherapy drugs but also increase the efficacy of combined chemoradiotherapy. In the future, the XPF–ERCC1 blocker may be used to improve the efficacy of 5-FU- and oxaliplatin-based CRT.

## 1. Introduction

In clinical cancer treatment, preoperative or postoperative adjuvant Chemoradiotherapy (CRT) is often used alongside surgery to improve patient outcomes [1,2,3]. In addition, in the treatment of colorectal cancer, the use of preoperative CRT was able to improve the anus retention rate and reduce tumor recurrence in patients with colorectal cancer [4,5,6,7]. In addition, chemoradiation is being administered as an organ preservation therapy combined with induction or consolidation chemotherapy (RAPDI OERA PRODIGE trials) more and more. However, our previous studies indicated that the high expression of Excision Repair Cross-Complementation Group 1 (ERCC1) contributes to lower efficiency of chemotherapy, poor response to radiotherapy, and poor prognosis in colorectal cancer [8,9,10]. In addition, it has been reported in many studies that the overexpression of ERCC1 reduces the overall- and event-free survival rate in non-small cell lung cancer, head and neck cancer, gastric cancer, esophageal cancer, and breast cancer undergoing CRT [11,12,13]. For example, in a study by Jun et al., patients with locally advanced squamous cell carcinoma of the head and neck treated with cisplatin-based concurrent chemoradiotherapy were collected for analysis of 3-year progression-free survival and overall survival rates and ERCC1 expression [14], the results suggested that ERCC1 expression might be a useful predictive marker of locally advanced squamous cell carcinoma of the head and neck in patients treated with cisplatin-based CRT. In addition, Kim et al. pointed out that the expression of ERCC1 can predict clinical outcomes after preoperative chemoradiotherapy in patients with localized esophageal cancer [15]. The ERCC1–XPF heterodimer plays an important role in DNA repair to facilitate several DNA repair pathways, including nucleotide excision repair, inter-strand cross-link repair, and double-strand break repair. The overexpression of ERCC1 in cancer patients might increase the effectiveness of chemotherapy and radiotherapy through the DNA repair of ERCC1. These research reports showed that the level of ERCC1 expression is a very important factor affecting the efficacy of radiotherapy and chemotherapy.

Therefore, ERCC1–XPF is an attractive target for drug design to enhance the therapeutic efficacy of chemotherapy [16,17]. For example, Li used siRNA to inhibit ERCC1 expression and was able to ameliorate resistance to chemotherapy in gastric carcinoma [13]. This proves that ERCC1 is a good target, but the clinical application of siRNA is still limited. Many studies also indicate that the small inhibitor of ERCC1 may provide a clinically applicable approach. For example, Elmenoufy and colleagues developed ERCC1 inhibitors to improve the sensitivity of UV radiation and cyclophosphamide in colorectal cells [18]. In addition, Jordheim and colleagues developed an XPF–ERCC1 blocker (NSC130813) that could suppress the interaction and function of the ERCC1–XPF complex to increase the in vitro therapeutic efficacy of mitomycin C [19]. These results suggest that the development of small molecules to suppress the function of ERCC1 can enhance the therapeutic efficacy of clinical drugs. However, this strategy may require further in vivo experiments to verify its feasibility.

Currently, 5-FU-based CRT and oxaliplatin (OXA)-based CRT are the approaches most often used to treat stage II and III CRC patients. Therefore, understanding whether XPF–ERCC1 blockers can improve the therapeutic efficacy of 5-FU-based CRT and OXA-based CRT in colorectal cancer is very important. To investigate whether the XPF–ERCC1 blocker can improve the therapeutic efficacy of 5-FU- and oxaliplatin-based CRT in colorectal cancer, we first analyzed the cytotoxicity of the 5-FU, OXA, and XPF–ERCC1 blocker to understand the dose-limitation of each compound. In addition, we also evaluated whether the ERCC1 inhibitor could improve the efficacy of 5-FU or OXA in colorectal cells. We also confirmed whether the XPF–ERCC1 blocker could increase the efficacy of 5-FU-based CRT or OXA-based CRT via inhibiting DNA repair. Finally, we used the HCT116 and SW620 tumor-bearing mice to analyze whether the XPF–ERCC1 blocker can increase the therapeutic effect of 5-FU, OXA, 5-FU-based CRT, or OXA-based CRT in vivo.

## 2. Materials and Methods

### 2.1. Cells Lines and Mice

HCT116 and SW620 colon cancer cell lines (ATCC, Manassas, VA, United States of America) were cultured in DMEM (Sigma-Aldrich, Burlington, MA, USA) supplemented with 10% of bovine calf serum, 100 units/mL of penicillin, and 100 μg/mL of streptomycin at 37 °C in an atmosphere of 5% CO_2_. Specific pathogen-free male BALB/c nude mice were obtained from the National Laboratory Animal Center, Taipei, Taiwan.

### 2.2. ERCC1 shRNA, XPF–ERCC1 Blocker (NSC130813), Chemotherapeutic Drug Treatment, Cell Viability Assay

ERCC1 shRNA was designed and constructed by GeneChem Co., Ltd. (Shanghai, China) and transfected by lipofectamine 2000 protocol (Thermo Fisher Scientific, Waltham, MA, USA). Cells were plated in a 96-well plate (2000/well) overnight and treated with serial dilution concentrations of NSC130813 (XPF–ERCC1 blocker, Sigma-Aldrich, Burlington, Massachusetts, USA) 5-FU (Haupt Pharma, Wolfratshausen, Germany) or oxaliplatin (Sanofi, Paris, France). Cell viabilities were detected by ATPlite kit (510-17281, PerkinElmer), and the luminescence value was measured by a multimode plate reader (PerkinElmer, Waltham, MA, USA), and isobologram plots were calculated using the Calcusyn software v.2.11 (Premier Biosoft International, San Jose, CA, USA).

### 2.3. Irradiation Expose and Colony Formation Assay

Cells (10^3^ cells/well) were exposed to ionizing radiation by a linear accelerator (Varian Clinac iX, Department of Radiation Oncology, Kaohsiung Medical University Hospital) before a bolus (1.5-cm-thick) was placed on the top and bottom of cell culture plates and cultivated for three weeks. The colonies were fixed with 5 mL of 10% neutral buffered formalin solution for 15–30 min and stained with 5 mL of 0.01% (*w*/*v*) crystal violet in dH_2_O for 30–60 min. Excess crystal violet was washed with dH_2_O, and dishes were allowed to dry. Colonies containing more than 50 individual cells were counted using a stereomicroscope.

### 2.4. Western Blotting

Protein extracts, separated by SDS-PAGE and transferred onto NC membranes (General Electric Company, Schenectady, NY, USA), were probed with antibodies against XPF (D3G8C, Rabbit mAb, #13465), Rabbit anti-γ-H2AX Recombinant mAb BLR053F (A700-053, 1:1000, Bethyl, Gurgaon, India), Anti-RAD 51 antibody (NBP2-20058, 1:1000, Bio-Techne, Minneapolis, MN, USA), or Anti-beta Actin antibody (mAbcam 8226, 1:1000, Abcam, Cambridge, UK). Proteins of interest were detected with peroxidase-conjugated AffiniPure Goat Anti-Mouse IgG, Fc Fragment Specific (115-035-008, 1:1000, Jackson ImmunoResearch, Bar Harbor, ME, USA), or goat anti-rabbit IgG, Fab’2-HRP (sc-3837, Santa Cruz Biotechnology, CA, USA) and visualized with the Immobilon Western Chemiluminescent HRP Substrate (WBKLS0500, Merck, Rahway, NJ, USA), according to the protocol provided. The quantification of WB bands was achieved through Image J (ImageJ bundled with 64-bit Java 8. https://imagej.nih.gov/ij/(accessed on 28 June 2022)).

### 2.5. Immunofluorescence Staining

Cells that were treated specify conditions and were fixed with 4% of paraformaldehyde containing 2% sucrose (573113, Merck, Rahway, NJ, USA) for 15 min and then washed with phosphate-buffered saline (PBS). Cells were then permeabilized with 0.25% Triton X-100 (CAS 9002-93-1, Triton X-100, USA) for 5 min, blocked with 1% of Bovine Serum Albumin (BSA, A3294-10G, Merck, Rahway, NJ, USA) for 1 h, and incubated with Rabbit anti-γ-H2AX Recombinant mAb BLR053F (A700-053, 1:1000, Bethyl, Gurgaon, India) in PBS containing 1% BSA overnight at 4 °C. Cells were then washed with PBS and incubated with FITC-conjugated secondary antibody for 2 h at room temperature. Cells were finally washed with phosphate-buffered Saline, 0.1% of Tween, and then stained for 5 min with DAPI (DAPI (4′,6-diamidino-2-phenylindole, Thermo Fisher Scientific, Waltham, MA, USA). The images were captured by a fluorescent microscope (Olympus Soft Imaging Solutions Gmbh, Hamburg, Germany). The image analysis was done by using Image J (ImageJ bundled with 64-bit Java 1.8.0_172, Wayne Rasband, Bethesda, Rockville, MD, USA, https://imagej.net/imagej-wiki-static/Downloads (accessed on 28 June 2022)).

### 2.6. Xenograft Animal Model, Chemotherapy Treatment Schedule and Chemoradiotherapy Treatment Schedule

HCT116 or SW620 cells (2 × 10^6^) were subcutaneously injected into specific pathogen-free male BALB/c nude mice. When tumors reached a size of approximately 50 mm^2^, PBS, 5-FU (5-Fu^®^), oxaliplatin (Eloxatin injection), XPF–ERCC1 blocker (NSC130183), or a combination of the above compounds were administered at 4 mg/kg/mouse by intraperitoneal injection every three days for one week. Radiation was performed with multiple doses of 2 Gy per fraction (cumulative radiation dose was 6 Gy). There were four mice in each group. The tumor size was measured by using a digital caliper (VWR International, Radnor, PA, USA), and the tumor volume was determined using the formula: tumor volume [mm^3^] = (length [mm]) × (width [mm])^2^ × 0.52. There are no special exclusion conditions for mice, and the mice were sacrificed on the 30th day. All the above were implemented in accordance with the regulations of the Experimental Animal Ethics Committee of Kaohsiung Medical University (IACUC: 110254).

### 2.7. Immunohistochemical Staining and Tunel Assay

All of the tumor tissues were fixed in 4% of formaldehyde for at least 6 h, embedded in paraffin, and then 4 μm-thick sections were cut. For immunohistochemical study slides, the sections were dried, deparaffinized, and then rehydrated. Antigen retrieval was performed using a heat-mediated Target Antigen Retrieval Buffer (pH 9.0; Dako, Glostrup, Denmark) for 10 min. The slides were washed with a Tris buffer solution after using 3% of hydrogen peroxide to block endogenous peroxidase activity for 10 min at room temperature. The sections were incubated with Rabbit anti-γ-H2AX Recombinant mAb BLR053F (A700-053, 1:500, Bethyl) for 2 h at room temperature. Tumor xenografts were surgically excised on day 30, placed in formalin, and embedded in paraffin. Tunel assay was conducted using an in situ Cell Death Detection Kit, POD (Lafayette, LA, USA) Cancers 2022, 14, 4053 4 of 20. All sections were scanned using the Olympus VS200 slide scanner (Olympus Soft Imaging Solutions Gmbh, Hamburg, Germany).

### 2.8. Statistical Analysis

Statistical significance was calculated using GraphPad Prism 6.0 with Student’s *t*-test and multiple *t*-tests. Data were considered statistically different at *p* < 0.05.

## 3. Results

### 3.1. XPF–ERCC1 Blockers Synergistically Enhance the Cytotoxicity of 5-FU or Oxaliplatin in Colorectal Cells

To investigate whether the XPF–ERCC1 blocker was able to improve the cytotoxicity efficacy of 5-FU- and oxaliplatin in colorectal cancer. We analyzed the cytotoxicity of the 5-FU, OXA, and XPF–ERCC1 blocker to understand the dose-limitation of each compound. HCT116 or SW620 were treated with 0.14, 0.41, 1.23, 3.7, 11.11, 33.33, and 100 μM of 5-FU, OXA, or the XPF–ERCC1 blocker. After 48 h, the cells were collected and the cell viability was analyzed. The results show that the cytotoxicity of the XPF–ERCC1 blocker (IC_50_ = 41.86 μM) was higher than that of 5-FU (IC_50_ = 29.10 μM) or OXA (IC_50_ = 2.62 μM) in HCT116 cells. The cytotoxicity of the XPF–ERCC1 blocker (IC_50_ = 60.85 μM) was higher than that of 5-FU (IC_50_ = 57.57 μM) or lower than OXA (IC_50_ = 69.39 μM) in the SW620 cells (Appendix A). In addition, we further clarified whether the incorporation of the XPF–ERCC1 blocker could increase the cytotoxicity of 5-FU and OXA in colorectal cells. HCT116 and SW620 were treated with different concentrations of 5-FU or OXA, and then the XPF–ERCC1 blocker (0.14, 0.41, 1.23, 3.7, 11.11 or 33.33 μM) was added into each group. After 48 h, the cells were collected, and the cell viability was evaluated to analyze the synergistic effect of the XPF–ERCC1 blocker. The results showed that the combined inhibitory effect of 5-FU with the XPF–ERCC1 blocker was 12.46 μM or 21.41 μM in HCT116 (Figure 1A) or SW620 (Figure 1B), respectively, and the combined inhibitory effect of OXA with the XPF–ERCC1 blocker was 1.46 μM or 17.76 μM in HCT116 (Figure 1C) or SW620 (Figure 1D), respectively. These results indicated that the incorporation of the XPF–ERCC1 blocker could increase the cytotoxicity of 5-FU and OXA in colorectal cells (Appendix A).

Furthermore, we further identified the drug synergy by CalcuSyn software. Representative examples of the graphs generated from the analysis of the XPF–ERCC1 blocker compound combinations demonstrate the synergistic effect of 5-FU or OXA in combination with the XPF–ERCC1 blocker in HCT116 and SW620 cells. These results suggest that combinational treatment with the XPF–ERCC1 blocker can synergistically enhance the therapeutic efficacy of 5-FU- and oxaliplatin-based CRT in colorectal cancer. Furthermore, we also calculated the CI (combination index) values of the merged XPF–ERCC1 blocker, 5-FU or OXA in HCT116 and SW620, and the results showed that their CI values were 0.72, 0.71, 0.58, and 1, respectively, when CI ≤ 1 means that he XPF–ERCC1 blocker could increase the toxicity of 5-FU and OXA to colorectal cancer cells. In addition, we also used the combination of high and low concentrations of XPF–ERCC1 blocker in serially diluted 5-FU and OXA, and the results showed that the XPF–ERCC1 blocker could increase the cytotoxicity of 5-FU and OXA (Appendix A).

### 3.2. XPF–ERCC1 Blocker Increases the Efficacy of 5-FU-Based CRT and OXA-Based CRT In Vitro

To further confirm that the XPF–ERCC1 blocker was able to increase the efficacy of OXA-based CRT and 5-FU-based CRT, HCT116 and SW620 were seeded in each well of 6-well dishes, respectively. The colony numbers were analyzed after being treated with 4 Gy, 0.4 μM of the XPF–ERCC1 blocker, 0.4 μM of 5-FU, 4Gy, and 0.4 μM of 5-FU combined, and 4Gy, 0.4 μM of 5-FU, and 0.4 μM of the XPF–ERCC1 blocker combined for two weeks. The results showed that the number of cell colonies in the radiation RT/5-FU/XPF–ERCC1 blocker group was lower than in the RT/5-FU group in HCT116 (Figure 2A,C) and SW620 (Figure 2B,D) cells. The same results were also found in OXA-based CRT. Therefore, the XPF–ERCC1 blocker was able to inhibit the ability of cell proliferation when the cell was treated with 5-FU-based CRT and OXA-based CRT. In addition, to further confirm whether the XPF–ERCC1 blocker could increase the cytotoxicity, cell viability was tested; the XPF–ERCC1 blocker enhanced the cytotoxicity of FU-based CRT and OXA-based CRT in HCT116 (Figure 2E,F) and SW620 (Figure 2G,H).

### 3.3. Inhibition of DNA Repair by XPF–ERCC1 Blocker Leads to Radio-Chemosensitization

To understand whether the XPF–ERCC1 blocker increases the cytotoxicity of radiation and chemotherapy drugs by inhibiting DNA repair, we seeded 10^5^ cells/well of HCT116 and SW620 in 6-well dishes and irradiated them with 4 Gy only, 0.4 μM of the XPF–ERCC1 blocker, 4 Gy, and 0.4 μM of 5-FU combined, or 4 Gy and 0.4 μM of OXA for 48 h. The cytoplasmic mass was collected, lysed, and the expression levels of XPF, γ-H2AX, and β-actin were detected. The results showed the levels of XPF in the radiation (RT) groups with 5-FU or OXA were higher than that in the control group, and the expression of XPF decreased when used in combination with the XPF–ERCC1 blocker. In addition, regarding the expression of γ-H2AX, the radiation and 5-FU group performed better than the control group, but the level of γ-H2AX significantly decreased after adding the XPF–ERCC1 blocker in HCT116 (Figure 3A,C). We also observed a similar phenomenon in SW620 (Figure 3B,D). These results imply that the sensitization of XPF–ERCC1 blocker to 5-FU or OXA chemoradiotherapy is caused by the inhibition of DNA repair. To understand the effect of the XPF–ERCC1 blocker on CRT sensitization, 5 × 10^4^ HCT116 and SW620 were seeded and treated with combined 0.4 μM of 5-FU, 0.4 μM of OXA, 4 Gy radiation or 0.4 μM of the XPF–ERCC1 blocker for 48 h. The level of γ-H2AX in the nucleus was detected by immunofluorescence staining. The results indicated that the expression levels of γ-H2AX in the CRT group with the XPF–ERCC1 blocker were higher than those without the XPF–ERCC1 blocker in HCT116 (Figure 3E,G) and SW620 (Figure 3F,H). This means that the XPF–ERCC1 blocker could increase the toxicity of the 5-FU base CRT and OXA base CRT by increasing DNA damage.

### 3.4. XPF–ERCC1 Blocker Increases Chemotherapeutic Efficacy In Vivo

To further understand whether the XPF–ERCC1 blocker could increase the chemotherapeutic efficacy of 5-FU and OXA in vivo, we implanted 2 × 10^6^ HCT116 or SW620 cells in the hind legs of mice. When the tumor volume grew to 50 mm^3^, PBS, 2 mg/kg of 5-FU, 2 mg/kg of the ERCC1–XPF blocker, and a combination of 2 mg/kg of 5-FU with 2 mg/kg of the ERCC1–XPF blocker was administered (Figure 4A). We then analyzed the effect of combining the XPF–ERCC1 blocker and 5-FU in the CRC animal model. The results showed that the average tumor volume of the XPF–ERCC1 blocker with the 5-FU group was less than the 5-FU only group in both HCT116 (624 mm^3^; 1011 mm^3^) and SW620 (742 mm^3^; 1671 mm^3^). Therefore, the combination with the XPF–ERCC1 blocker was able to increase the chemotherapeutic efficacy of 5-FU in HCT116 (Figure 4B) and SW620 (Figure 4C). Furthermore, we also analyzed PBS, 2 mg/kg of OXA, 2 mg/kg of the XPF–ERCC1 blocker, and a combination of 2 mg/kg of OXA and 2 mg/kg of the ERCC1–XPF blocker. The tumor volume was measured up to the 30th day.

The results indicated the average tumor volume of the combined XPF–ERCC1 blocker and 5-FU group was less than the OXA-only group in HCT116 (412 mm^3^; 593 mm^3^) or SW620 (1114 mm^3^; 2218 mm^3^) cells. These results demonstrate that the 2 mg/kg of OXA with 2 mg/kg of the XPF–ERCC1 blocker group had higher tumor limits than that of the OXA-only group. Therefore, the XPF–ERCC1 blocker also enhanced the chemotherapeutic efficacy of OXA in HCT116 (Figure 4D) and SW620 (Figure 4E)

### 3.5. XPF–ERCC1 Blocker Increases the Efficacy of Combined Radiochemotherapy In Vivo

To further understand whether the XPF–ERCC1 blocker can increase CRT efficacy, we implanted 2 × 10^6^ HCT116 or SW620 into the hind legs of mice, and when the tumor volume reached 50 mm^3^, they were given 6 Gy of radiation therapy (2 Gy/time, total cumulative 6 Gy), or 2 mg/kg of 5-FU and 6 Gy, 2 mg/kg of 5-FU and 6 Gy of radiation with 2 mg/kg of the XPF–ERCC1 blocker, or 2 mg/kg 5-FU, the XPF–ERCC1 blocker, and 6 Gy radiation, and the tumor volume was measured until the 30th day (Figure 5A). The mean tumor volume of the 5-FU-based CRT group combined with the XPF–ERCC1 blocker was smaller than that of the 5-FU-based CRT group (561 mm^3^,446 mm^3^) in the HCT116 tumor (Figure 5B). Similar results were also found in SW620 (705 mm^3^, 247 mm^3^ Figure 5C). Furthermore, the mean tumor volume of the OXA-based CRT group combined with XPF–ERCC1 blocker was smaller than that of the OXA-based CRT group (356 mm^3^,1121 mm^3^) in the HCT116 tumor (Figure 5D). Similar results were seen in SW620 tumors (487 mm^3^, 601 mm^3^). Therefore, XPF–ERCC1 blockers were able to increase the efficacy of 5-FU-based CRT and OXA-based CRT in HCT116 and SW620 tumors (Figure 5E), which indicates that the combined use of the XPF–ERCC1 blocker not only improves the efficacy of OXA or 5-FU but also increases the combination of OXA or 5-FU radiation therapy. To confirm the DNA damage in tumor tissues, we analyzed the levels of r-H2AX and apoptosis in the tumor mass through an IHC and Tunel assay. The results indicate that the group combining the XPF–ERCC1 blocker and radiation irradiation has a higher staining area and a more scattered interior of the tumor tissue. It is suggested to combine XPF–ERCC1 blockers that could enhance DNA damage to improve the therapeutic efficacy of 5-FU-based CRT or OXA-based CRT in HCT116 (Figure 5F,H) and SW620 (Figure 5G,I).

## 4. Discussion

In this study, we confirmed that the incorporation of the XPF–ERCC1 blocker could increase the cytotoxicity of 5-FU and OXA in colorectal cells. Furthermore, we also found that the 5-FU-based CRT group or OXA-based CRT group combined with the XPF–ERCC1 blocker could reduce the XPF level and increase the expression of γ-H2AX in colorectal cancer cells suggesting that the XPF–ERCC1 blocker could increase the cytotoxicity of radiation and chemotherapy drugs by inhibiting the XPF–ERCC1-based DNA repair. Furthermore, we used the tumor-bearing mice to confirm that the XPF–ERCC1 blocker enhanced the therapeutic effect of 5-FU or OXA; it also improved the 5-FU-based CRT or OXA-based CRT therapy for RC. Therefore, we believe that XPF–ERCC1 blockers could be used as a sensitizer of chemoradiotherapy to improve the prognosis of CRC patients. However, NSC130813 was also demonstrated to inhibit HSPA5. It may be the result of off-target effects. Therefore, we also confirmed it enhanced the effect of chemotherapy and radiation therapy by silencing ERCC1 in colorectal cancer cells. While the above studies have confirmed that ERCC1–XPF inhibitors could increase the efficacy of 5-FU, OXA, and radiation, it needs to be further verified by expanding the sample size. In addition, the pharmacokinetics of the ERCC1–XPF inhibitor also needs to be confirmed, and cancers should be treated with CCRT such as colorectal cancer in the future.

Suppressing the function of ERCC1–XPF is very important to increase the effectiveness of chemotherapy and radiotherapy in cancer because both chemotherapy and radiotherapy cause cancer cell death through DNA damage, DNA cross-link, and DNA break. Therefore, under chemotherapy or radiotherapy, ERCC1–XPF heterodimer plays an important role in DNA repair to maintain cell survival. For example, Liu et al. pointed out that ERCC1 can affect the therapeutic effect of 5-FU in human gastric cancer cells [20]. Furthermore, in Faridounnia et al., it was pointed out that ERCC1 can reduce DNA damage by removing a platinum chemo-drug complex formation with DNA in many cancers. [21]. In our previous study, we also reported that the overexpression of ERCC1 leads to resistance to radiation in colorectal cancer [22]. In fact, the ERCC1–XPF heterodimer catalyzes the 5′ nick in the process of excising DNA damage to facilitate several DNA repair pathways, including nucleotide excision repair, inter-strand cross-link repair, and double-strand break repair [23,24,25,26]. For example, Ferry et al. reported that ERCC1–XPF endonuclease is a determinant of increased nucleotide excision repair in cisplatin-resistant ovarian cancer cells [27]. Similar studies reported that ERCC1-associated nucleotide excision repair pathways are involved in cisplatin resistance in non-small-cell lung cancer [28]. We also found the 5-FU-based CRT group or the OXA-based CRT group combined with the XPF–ERCC1 blocker could reduce the XPF level and increase the expression of γ-H2AX in colorectal cells (Figure 4), suggesting that the XPF–ERCC1 blocker was able to increase the cytotoxicity of radiation and chemical drugs by inhibiting the XPF–ERCC1 based DNA repair. Therefore, the inhibition of the XPF–ERCC1 can inhibit the DNA repair pathway, including nucleotide excision repair, interstrand cross-link repair, and double-strand break repair. This inhibition of XPF–ERCC1 can increase the efficacy of chemotherapy and radiotherapy in cancer.

We suggest that the XPF–ERCC1 blocker may be used widely in the treatment of many different cancers. Our study indicated that the XPF–ERCC1 blocker increased the therapeutic effect of 5-FU, OXA, or RT. Combining the XPF–ERCC1 blocker with 5-FU or OXA enhanced the cytotoxicity of the chemotherapy drugs in colorectal cells. Furthermore, the XPF–ERCC1 blocker also increased the cytotoxicity of 5-FU-based CRT and OXA-based CRT by inhibiting the XPF product in the DNA locus. In vivo, the XPF–ERCC1 blocker was confirmed to enhance the therapeutic efficacy of 5-FU, OXA, 5-FU-based CRT, and OXA CRT. Furthermore, 5-FU is an antimetabolite drug that has been widely used to treat different types of cancer, including lung cancer, stomach cancer, and RC. For example, Blondy et al. reported that 5-FU was used in RC of local or distant invasion [29]. 5-FU induces cytotoxicity either by interfering with essential biosynthetic activity by inhibiting the action of thymidylate synthase or by misincorporating its metabolites into DNA [30,31,32]. Similar to FU, OXA is also a drug used in the clinical treatment of recurrent metastatic gastric cancer. In addition, OXA is also a third-generation platinum compound and a U.S. FDA-approved first-line chemotherapy treatment for CRC [33]. In cancer cells, OXA binds preferentially to the guanine and cytosine moieties of DNA, leading to cross-linking of DNA, thus inhibiting DNA synthesis and transcription to cause apoptosis [34,35]. Finally, radiation is often used in clinical combination therapy for breast cancer, head and neck cancer, and other cancers [36], and its mechanism is also through the destruction of DNA in cancer cells through apoptosis. In summary, whether based on 5-FU, OXA, or RT for cancer adaptation, or mechanism of action, we re-emphasize that the XPF–ERCC1 blocker has potential as a therapeutic sensitizer in many different types of cancers.

The research of Claudia Weilbeer and his colleagues considers the absence of DNA damage and ERCC1 and XPF in the absence of heterodimerization. They would not localize to the nucleus of damaged cells [19]. Therefore, ERCC1 is a safe target with low side effects. The side effects of small molecule drugs mainly come from on-target toxicity and off-target toxicity. ERCC1 is a protein that is expressed at low levels in normal cells and abundantly expressed when DNA is damaged. For example, in the clinical study by Jia et al., it was reported that ovarian cancer cells had higher expression of ERCC1 than normal tissue or normal ovaries [37]. In addition, in the treatment environment of radiation and chemotherapy drugs, ERCC1 was produced via cancer cells immediately. For example, in Wang et al. it was reported that non-small cell lung cancer can cause a large amount of ERCC1 expression after chemotherapy [38]. Liu et al. also pointed out that ERCC1 expression can also be induced by chemotherapy in ovarian cancer [21]. Furthermore, in Xie et al., it was also proposed that radiation induces a large amount of ERCC1 expression in colorectal cancer stem-like cells [22]. The above examples all show that the expression of ERCC1 increases in cancer cells after radiochemotherapy. Through the above evidence, we know that the basic value of ERCC1 expression in cancer cells is higher than in normal cells, and a high level of expression can be induced by chemotherapy drugs or radiation. Therefore, using ERCC1 as a sensitive target has few side effects and is safe. If the safety of the drug is to be further improved, a targeted drug delivery system will be needed to increase the selectivity of the drug in cancer cells and further reduce its toxicity [23,39]. Based on the above discussion, we propose that this inhibitor has the following advantages: (1) suppression of the function of ERCC1–XPF, which is very important to increase the chemotherapy and radiotherapy in cancer; (2) the XPF–ERCC1 blocker may be used widely in the treatment of many different cancers; (3) ERCC1 has low toxicity and is a safe target. Therefore, in the future, the development of the XPF–ERCC1 blocker might improve the quality of life of cancer patients receiving radiation or chemotherapy drugs.

## Figures and Tables

**Figure 1 cells-12-01475-f001:**
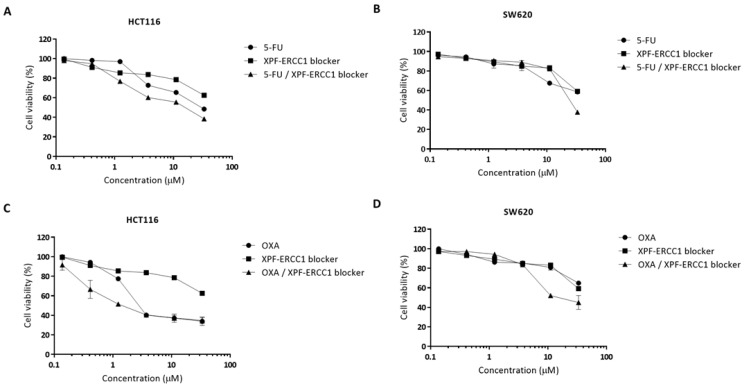
Combined inhibitory effect of 5-FU, OXA, and the XPF–ERCC1 blocker on cell growth in colorectal cancer cells. (**A**,**C**) HCT116 and (**B**,**D**) SW620 cells were treated with 5-FU, OXA, the XPF–ERCC1 blockers, the 5-FU/XPF–ERCC1 blocker, or the oxaliplatin/XPF–ERCC1 blocker at a constant ratio of 5-FU: XPF–ERCC1 blocker, 1:1 (HCT116 cells) and 1:1 (SW620 cells) for 72 h at different concentrations. Cell growth inhibition was determined by an ATP lite cell proliferation assay. Cell viability (%) = Sample Absorbance/Control Absorbance × 100. IC50 was calculated by Prismed. IC50, half-maximal inhibitory concentration. Data are presented as the mean + standard deviation. All experiments were independently repeated four times.

**Figure 2 cells-12-01475-f002:**
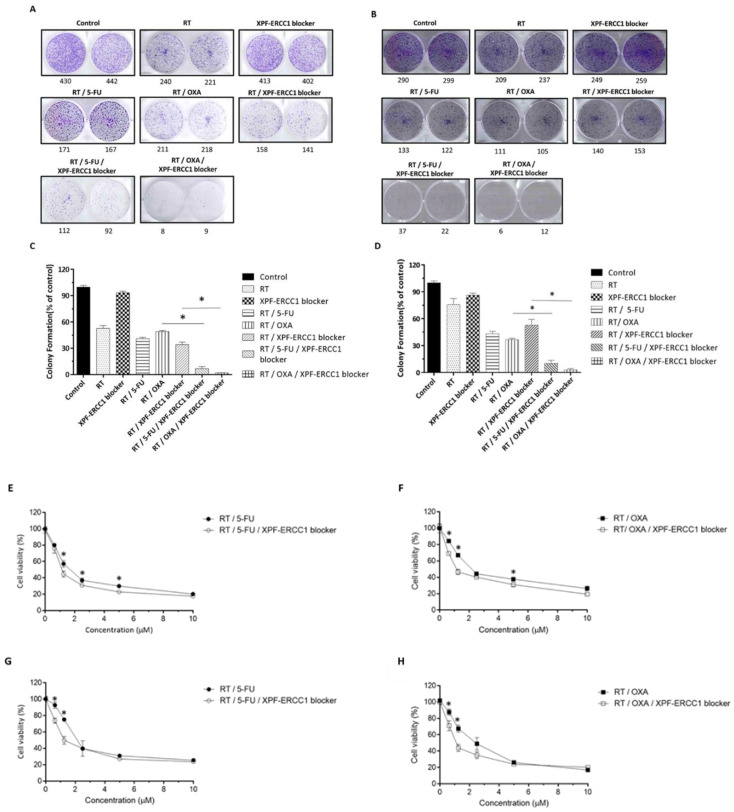
The XPF–ERCC1 blocker increases the efficacy of 5-FU-based CRT and OXA-based CRT in vitro. HCT116 (**A**) and SW620 (**C**) were treated with 4 Gy (RT), 0.4 μM of the XPF–ERCC1 blocker combined with 4 Gy, 0.4 μM of 5-FU (RT/5-FU) combined with 4 Gy, 0.4 μM of OXA (RT/OXA) combined with 4 Gy, 0.4 μM of 5-FU and 0.4 μM of the XPF–ERCC1 blocker (RT/5-FU/XPF–ERCC1 blocker) combined with 4 Gy, 0.4 μM of OXA and 0.4 μM of the XPF–ERCC1 blocker (RT/OXA/XPF–ERCC1 blocker). After 14 days, all cells were analyzed by counting the colony numbers. The colony formation (% of control) was calculated as sample colony numbers/control colony numbers (**B**,**D**). HCT116 (**E**,**F**) and SW620 (**G**,**H**) were treated with 4 Gy (RT) and combined with the sequence diluted 5-FU, OXA, and the XPF–ERCC1 blocker. After 48 h, analysis of cell viability was achieved via ATP Lite. Data are presented as the mean + standard deviation (*n* = 4; * *p* < 0.05). All experiments were independently repeated four times. Data are presented as the mean + standard deviation (*n*  =  4; * *p* < 0.05). All experiments were independently repeated four times.

**Figure 3 cells-12-01475-f003:**
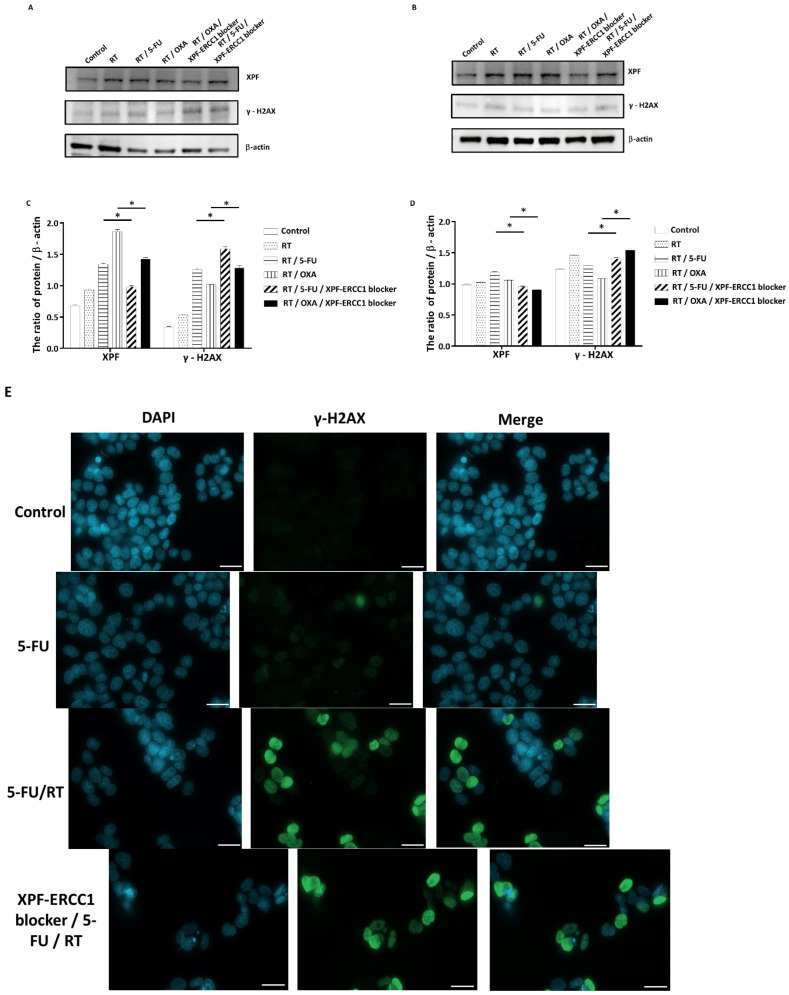
The XPF–ERCC1 blocker increases the effect of chemoradiotherapy on DNA damage. HCT116 (**A**) or SW620 (**B**) cells treated with RT, RT/5-FU, RT/OXA, RT/5-FU/XPF–ERCC1 blocker, and RT/OXA/XPF–ERCC1 blocker were analyzed by anti-XPF mAb, anti-γ-H2AX mAb, and anti-β-actin mAb after 48 h. Quantification and statistical analysis of HCT116 (**C**) and SW620 (**D**). The images of immunofluorescent staining for γH2AX in HCT116 (**E**,**G**) and SW620 (**F**,**H**) upon treatment with 5-FU (0.4 μM), 5-FU based CRT (0.4 μM 5-FU/4 Gy radiation), combined XPF–ERCC1 blocker (0.4 μM) with 5-FU based CRT, OXA (0.4 μM), OXA based CRT (0.4 μM OXA/4 Gy radiation) or in the XPF–ERCC1 blocker combination. Scale bar, 10 μm. Data are presented as the mean + standard deviation (*n* = 4; * *p* < 0.05). All experiments were independently repeated four times.

**Figure 4 cells-12-01475-f004:**
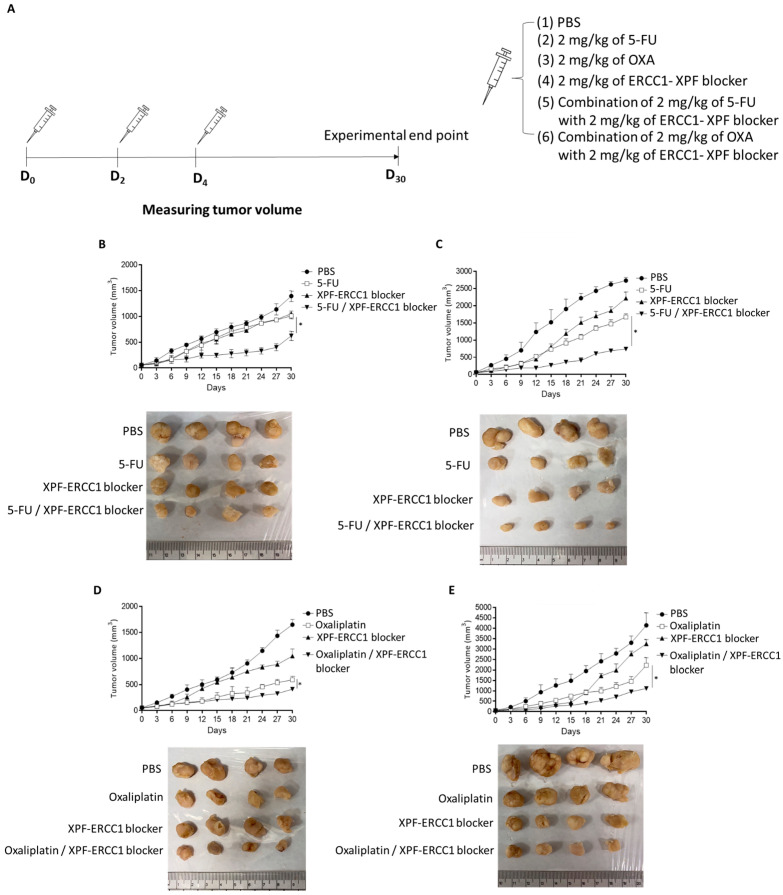
The XPF–ERCC1 blocker improved the in vivo chemotherapeutic efficiency of 5-FU and oxaliplatin. HCT116 or SW620 cells (2 × 10^6^) were subcutaneously injected into specific pathogen-free male BALB/c nude mice. When tumors reached a size of approximately 50 mm^2^ at D_0_, PBS, 5-FU, oxaliplatin, the XPF–ERCC1 blocker, or a combination of the above compounds were administered at 4 mg/kg/mouse simultaneously by intraperitoneal injection every three days for one week and measured tumor volume until to D_30_ (**A**). A xenograft model was divided into treatment four groups: PBS, 5-FU, Oxaliplatin, the XPF–ERCC1 blocker, the 5-FU/XPF–ERCC1 blocker, and the Oxaliplatin/XPF–ERCC1 blocker. Tumor volumes were measured between days 0 and 30 in HCT116 (**B**,**D**) and SW620 (**C**,**E**). Data are shown as subfigures (*p* < 0.05 *). All experiments were independently repeated four times.

**Figure 5 cells-12-01475-f005:**
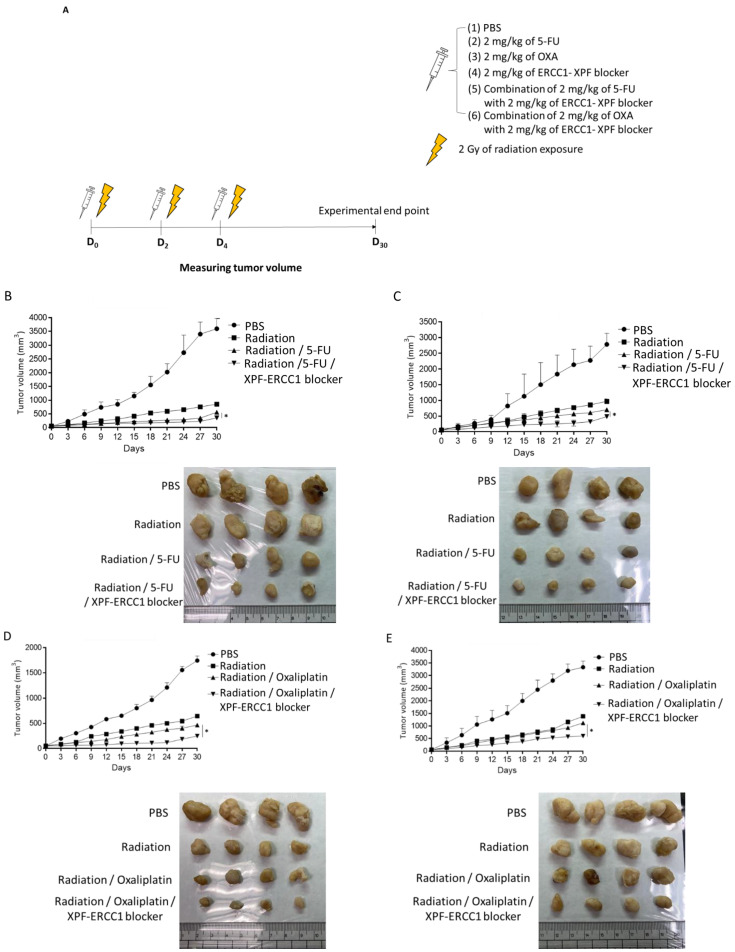
The XPF–ERCC1 blocker enhances the in vivo chemo-radiotherapeutic efficiency. HCT116 or SW620 cells (2 × 10^6^) were subcutaneously injected into specific pathogen-free male BALB/c nude mice. When tumors reached a size of approximately 50 mm^2^ at D0, PBS, 5-FU, oxaliplatin, the XPF–ERCC1 blocker, 2 Gy radiation expose (total 6 Gy) or a combination of the above compounds were administered at 4 mg/kg/mouse simultaneously by intraperitoneal injection every three days for one week and measured tumor volume until D_30_ (**A**). A xenograft model was divided into six treatment groups: Radiation, Radiation/5-FU, Oxaliplatin, the XPF–ERCC1 blocker, the 5-FU/XPF–ERCC1 blocker, and the Oxaliplatin/XPF–ERCC1 blocker. Tumor volumes were measured between days 0 and 30 in HCT116 (**B**,**D**) and SW620 (**C**,**E**) cells. The mice were sacrificed on the 30th day, and the tumor tissues were detected with a γ-H2AX level and apoptosis level by a specific mAb and Tunel assay in HCT116 (**F**,**G**) and SW620 (**H**,**I**), the positive staining area was indicated by the arrow. Data are shown as subfigures (*p* < 0.05 *). All experiments were independently repeated four times.

## Data Availability

The data analyzed during the current study are available from the corresponding author upon reasonable request.

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
