# Peer review of "XPF–ERCC1 Blocker Improves the Therapeutic Efficacy of 5-FU- and Oxaliplatin-Based Chemoradiotherapy in Colorectal Cancer"

_cells, 2023, doi:10.3390/cells12111475_

Round 1
Reviewer 1 Report
The authors have performed an invitro and animal study evaluating the addition of a novel ERCC1 blocking agent to 5-FU or oxaliplatin based chemoradiation. The study is well done and written . especially the combination of the cell lines and the mice model as well as the confirmation that silencing the ERCC1 gene provides similar result give a convincing set of data to conclude that ERCC1 blockage might ve interesting to explore infase1 and 2 trials.
Some comments from a clinical view
THE introduction mentions that ERCC1 overexpression is frequently seen and is associated with poor prognosis . However on the other hand ERCC1 protein expression is reduced or absent in 84% to 100% of colorectal cancers and lower expression of ERCC1 has been reported as being associated with unfavorable prognosis in patients undergoing treatment with oxaliplatin. This is conflicting with current introduction. Is true that during radiation ERCC1 is overexpressed a reaction to damage and this might be a more convincing reasoning to explore the ERCC1 blockage as enhancer of the chemoradiation.
Introduction; chemoradiation is more and more being administered as organ preservation therapy combined with induction or consolidation chemotherapy (RAPDI OERA PRODIGE trials)
Potential side effects ; absence of ERCC1 is not viable for human and mice; long term usage should introduce side effects such as melanoma’s by UV radiation, mucosal damage etc
HCT 116 and SW620 are coloncancer cell lines ; not rectal cells as frequently mentioned
Figure 1 and 2 should be combined to clearly demonstrate the attributive effect of XPF-ERCC1 bl
What is the human plasma concentration of OX, 5FU and XPF –ERCC1 ‘is it comparable with mice experiment and cell cultures
Author Response
Dear Reviewer 1: please see the attachment

Reviewer 2 Report
Authors present a work on the XPF-ERCC1 blocking as a strategy to improve rectal cancer outcome. However, some points need to be addressed.
- some relevant references are missing in the introduction relative to the role of chemotherapy and TNT (PMID: 33287114, 29407455, 33301740, 33862000)
- some titles are vague and must be more adherent to what authors mean, for instance the first results paragraph where authors state "enhance therapy"
- some starting sentences are not actual sentences: for instance lines 159-160 and others all along the manuscript
- fig. 1 must be a single figure, this improves the readability of the figure, the same for fig. 2
- fig. 3 can be moved to supplementary as it does not add anything
- authors must add more details on the combination treatment: doses and times
- how did authors plan this combination treatment? contemporary treatment, starting with XPF-ERCC1 inhibition or with platinum? this is important as it is supposed that ERCC1 is engaged in platinum damage repair. If not performed, they must perform experiments with different treatment sequences
- why did authors perform colony forming assay only to study the combination effect for CRT? colony forming assays must be performed for 5FU and platinum combinations, and also for radiotherapy. On the other hand, proper cytotoxicity assays must be performed for CRT.
- besides gammaH2AX foci, RAD51 foci must be analyzed, adding if possible another assay such as immunofluorescence
- the induction of DNA damage must be validated in in vivo experiments
Author Response
Dear Reviewer 2: please see the attachment

Round 2
Reviewer 2 Report
Authors have properly addressed all reviewer's suggestions. No further comments.